# Integrative genomic characterization of five *Pediococcus acidilactici* strains reveals differing probiotic safety profiles

Yuda Disastra[1,2], Thidathip Wongsurawat[3], Piroon Jenjaroenpun[3], David John Hampson[4], Ratchnida Kamwa[1], Nuvee Prapasarakul[2,5]*

1 Veterinary Science and Technology, Faculty of Veterinary Science, Chulalongkorn University, Bangkok, Thailand, 2 Department of Veterinary Microbiology, Faculty of Veterinary Science, Chulalongkorn University, Bangkok, Thailand, 3 Faculty of Medicine, Siriraj Hospital, Mahidol University, Bangkok, Thailand, 4 School of Veterinary Medicine, Murdoch University, Perth, Western Australia, Australia, 5 Center of Excellence in Diagnosis and Monitoring Animal Pathogens (DMAP), Faculty of Veterinary Science, Chulalongkorn University, Bangkok, Thailand

* Nuvee.p@chula.ac.th

## Abstract

The increasing use of probiotics in livestock necessitates rigorous safety assessments to mitigate risks such as their inadvertent contribution to antimicrobial resistance (AMR) and horizontal gene transfer (HGT). This study employs whole-genome sequencing using both long-read (GridION, Oxford Nanopore Technologies) and short-read (Illumina, San Diego, CA, USA) platforms to assess the genomic and plasmidome profiles of five Thai strains of *Pediococcus acidilactici*, that previously have been evaluated for probiotic potential in livestock. Our comprehensive analysis identified genes encoding AMR, virulence factors, and probiotic-related genes. Notably, strains AF2519 and AF2019 harbored plasmid-borne *tet(M)* and *erm(B)* genes, with *tet(M)* embedded in a novel composite genetic arrangement flanked by mobile elements, suggesting historical recombination and altered mobility potential. Strains IAF6519, IAF5919, and P72N, free from plasmid-borne AMR genes, emerged as safer candidates, lacking virulence genes. Phenotypic tests revealed discrepancies with genomic data; for instance, AF2019 was resistant to clindamycin without detectable genes, and showed susceptibility to tetracycline despite the presence of *tet(M)*. The absence of complete transfer machinery in AF2519 and AF2019 suggests a reduced HGT risk. These findings underscore the importance of integrating genomic and phenotypic approaches in probiotic safety evaluations. The presence of plasmid-borne AMR genes in certain strains advises caution in their use, impacting probiotic selection and regulatory compliance in agriculture. This research informs policies and best practices for safe probiotic deployment, ensuring both efficacy and safety.

purpose. The work is made available under the Creative Commons CC0 public domain dedication.

**Data availability statement:** Sequence data supporting this study are available in the National Center for Biotechnology Information (NCBI) under BioProject accession number PRJNA1037750 (https://www.ncbi.nlm.nih.gov/bioproject/PRJNA1037750)

**Funding:** The author(s) received no specific funding for this work.

**Competing interests:** The authors have declared that no competing interests exist.

**Abbreviations:** ANI, Average Nucleotide Identity; BLAST, Basic Local Alignment Search Tool; CARD, Comprehensive Antibiotic Resistance Database; CDS, Coding Sequences; CLSI, Clinical and Laboratory Standards Institute; EFSA, European Food Safety Authority; EMBL, European Molecular Biology Laboratory; GECs, Genetic Exchange Communities; HCCA, α-cyano-4-hydroxycinnamic acid; HGT, Horizontal Gene Transfer; IBC, Institutional Biosafety Committee; iTOL, Interactive Tree of Life; KEGG, Kyoto Encyclopedia of Genes and Genomes; LAB, Lactic Acid Bacteria; MALDI-TOF MS, Matrix-Assisted Laser Desorption/Ionization-Time of Flight Mass Spectrometry; MGEs, Mobile Genetic Elements; MICs, Minimum Inhibitory Concentrations; MRS, deMan Rogosa and Sharpe; NCBI, National Center for Biotechnology Information; OriT, Origin of Transfer; RAST, Rapid Annotation using Subsystem Technology; rRNA, Ribosomal RNA; SNPs, Single Nucleotide Polymorphisms; VFDB, Virulence Factor Database; WGS, Whole Genome Sequencing; μl, Microliter

## Introduction

Probiotics, defined by the Food and Agriculture Organization of the United Nations and the World Health Organization [1] as live microorganisms that confer health benefits when administered in adequate amounts, are increasingly sought for inclusion in livestock feed due to their benefits in gut health maintenance, immune modulation, and prevention of gastrointestinal disorders [2,3]. Among the diverse groups of lactic acid bacteria (LAB), *Pediococcus acidilactici* is notable for its robust probiotic properties, including antimicrobial activity and enhancement of gut health, which are crucial for improving livestock performance and health [4,5]. Despite these attributes, detailed genomic characterization of *P. acidilactici* strains is essential to fully assess their safety and efficacy.

Traditional phenotypic evaluations of probiotic strains often fail to capture genetic diversity and potential safety concerns such as their antimicrobial resistance profiles and presence of mobile genetic elements (MGEs) [6,7]. Whole genome sequencing provides a comprehensive approach to uncover these genetic traits, thereby supporting safer and more effective probiotic applications [8]. The European Food Safety Authority (EFSA) also underscores the importance of genetic investigations to identify antibiotic-resistant genes, especially those associated with MGEs, to mitigate health risks and prevent the spread of resistance [9,10].

This study focuses on five Thai strains of *P. acidilactici*—IAF6519, IAF5919, AF2519, AF2019 and P72N that were isolated from chicken and pig feces [11]. These strains have been shortlisted as potential probiotics due to their resilience under gastrointestinal conditions and demonstrated antagonistic effects against enteric pathogens including *Escherichia coli* and *Salmonella* [12]. By employing whole genomic sequencing (WGS), the study aimed to provide a comprehensive characterization of these strains, enhancing understanding of their functional properties and safety profiles, with broader implications for animal nutrition and functional food development.

## Materials and methods

### Bacterial strains and growth conditions

This study utilized five Thai strains of LAB, specifically *P. acidilactici*, obtained from the culture collection at the Faculty of Veterinary Science, Chulalongkorn University, Thailand. The strains included IAF6519, IAF5919, AF2519, AF2019 and P72N, all isolated from fecal samples of chickens or pigs. Strains IAF6519, IAF5919, AF2519, and AF2019 were sourced from various types of chickens, including broilers, layers, and native Thai chickens—across both commercial and household farms [11,13]. Strain P72N was isolated from a pig fecal sample collected from a farm that did not administer antibiotics. All strains were preserved in deMan Rogosa and Sharpe (MRS) broth supplemented with 20% glycerol and stored at −20°C. Prior to experimental use, the strains were cultured in MRS broth at 37°C for 24 hours. Following this, the cultures were plated on MRS agar and incubated for an additional 48 hours. All procedures adhered to institutional guidelines, with the use of LAB strains approved by the Chulalongkorn University Institutional Biosafety Committee (IBC) (Approval Number: 2,331,008)

## Species confirmation of *P. acidilactici* strains

The species identity of the *P. acidilactici* strains was confirmed using two methods: Matrix-Assisted Laser Desorption/Ionization-Time of Flight Mass Spectrometry (MALDI-TOF MS) and the VITEK 2 Compact system. For MALDI-TOF MS analysis, colonies from MRS agar were placed on a target plate and overlaid with a solution of formic acid and α-cyano-4-hydroxycinnamic acid (HCCA). The analysis was conducted using MALDI Biotyper software (Bruker Daltonics), with species identification confidence assessed by score values: scores >2.3 indicated high confidence, 2.3–2.0 suggested high probability of correct species identification, and 2.0–1.7 indicated genus identification [14]. Following MALDI-TOF MS analysis, the VITEK 2 Compact system was employed for further confirmation. Colonies were suspended in 0.45% NaCl to achieve a McFarland standard of 0.50–0.63. The samples were then processed using VITEK GP ID and CBC cassettes at 35°C for six hours [15].

## Antibiotic susceptibility testing of *P. acidilactici* strains

Antibiotic susceptibility of the strains was evaluated using the broth microdilution method following the guidelines of the Clinical and Laboratory Standards Institute (CLSI). The antibiotics tested included: ampicillin, vancomycin, gentamicin, kanamycin, streptomycin, erythromycin, clindamycin, tetracycline, and chloramphenicol. The test was performed using THAPH and THA1CLF Sensititre plates (Thermo Fisher Scientific), and the minimum inhibitory concentrations (MICs) were determined using the Sensititre Vizion Digital MIC Viewing System. Results were compared to microbiological cut-off values established by the EFSA to classify strains as either resistant (R) or susceptible (S). To ensure accuracy, two quality control strains, *Enterococcus faecalis* ATCC 29,212 and *Escherichia coli* ATCC 25,922 were included in each assay to validate MIC readings [16].

## Hemolytic activity testing of *P. acidilactici* strains

The hemolytic activity of the LAB strains was assessed on tryptic soy agar (TSA) plates supplemented with 5% (v/v) sheep blood. After 48 hours of incubation at 37°C, the plates were examined for zones of hemolysis around bacterial colonies [17]. Three types of hemolysis were considered in this study. Beta-hemolysis (clear zone indicating complete red blood cell lysis), alpha-hemolysis (greenish zone indicating partial lysis), and gamma-hemolysis (no clear or greenish zone indicating no hemolytic activity) [18].

## Whole-genome sequencing and analysis of *P. acidilactici* strains

DNA from the five Thai strains was extracted using the ZymoBIOMICS DNA Miniprep Kit [19]. Quality and quantity were assessed with a NanoDrop spectrophotometer and Qubit fluorometer (Life Technologies, Carlsbad, CA, USA) [20]. Following extraction, sequencing was performed using long-read (GridION, Oxford Nanopore Technologies) and short-read (Illumina , San Diego, CA, USA) technologies.

Raw nanopore reads were processed with Nanoplot v1.33.0 [21], and filtered FASTQ files were assembled using Flye v2.9.1-b1780 [22]. The assembled genomes were polished with NextPolish [23] and plasmid assembly was conducted using Unicycler [24]. Genome quality was assessed with QUAST v5.0.2 and CheckM v1.1.9 [25], and plasmid typing was performed using MOB-suite v3.0.3 [26,27]. The bacterial chromosome was identified by aligning against the GTDB-Tk database (R207_v2), and functional genes were annotated using Prokka v1.14.6 [28]. The genome was visualized with PyCircos and Proksee [29].

All analyses were conducted at the Siriraj Long-Read Lab (Si-LoL), Faculty of Medicine Siriraj Hospital, Mahidol University, Bangkok, Thailand. Sequence data supporting this study are available in the National Center for Biotechnology Information (NCBI) under BioProject accession number PRJNA1037750 (https://www.ncbi.nlm.nih.gov/bioproject/PRJNA1037750) (S1 Table).

## Phylogenetic analysis of *P. acidilactici* strains

Identification of the Thai LAB strains was performed using Average Nucleotide Identity (ANI) analysis with GTDB-Tk v2.1.0 [30,31,32]. For orthologous ANI analysis, the OrthoANI Tool (OAT) from CJ Bioscience on the EzBioCloud server was utilized [33]. Species-level predictions were made using SpeciesFinder v.2.0 based on genomic data [34]. To elucidate evolutionary relationships among the strains, a maximum-likelihood phylogenetic tree was constructed using core single nucleotide polymorphisms (SNPs) with RAxML [35], incorporating genomic sequences from the five Thai strains and 15 reference strains of *P. acidilactici* obtained from the NCBI database (S1 Table). The resulting phylogenetic tree was visualized and analyzed using the Interactive Tree of Life (iTOL) tool from the European Molecular Biology Laboratory (EMBL) [36]. Additionally, protein-encoding sequences were analyzed to identify the pan- and core-genomes of the five Thai *P. acidilactici* strains and *P. acidilactici* HN9 (from traditional Thai-style fermented beef Nhang). This was accomplished using the Orthovenn web platform (https://orthovenn3.bioinfotoolkits.net) with default parameters [37].

## Analysis of antimicrobial resistance genes and virulence factors in *P. acidilactici*

An analysis of AMR genes and virulence factors was conducted on the genomes of the five Thai *P. acidilactici* strains using several publicly available databases. The Resistance Gene Identifier (RGI, v6.0.3) from the Comprehensive Antibiotic Resistance Database (CARD, v3.2.8), available at https://card.mcmaster.ca/analyze/rgi [38], ResFinder (v4.6) available at http://genepi.food.dtu.dk/resfinder [39], and the BlastKOALA tool from the KEGG database were employed for AMR gene identification, specifically inspected under "Brite ko01504: Antimicrobial resistance genes" [40]. For virulence factors, the Virulence Factor Database (VFDB) was utilized, accessible at http://www.mgc.ac.cn/cgi-bin/VFs/v5/main.cgi [41]. The analysis applied stringent criteria, requiring coverage and identity greater than 80% and an E-value of less than 1e-10 to ensure reliable results. This comprehensive approach enabled a robust evaluation of the genetic determinants associated with AMR and virulence, providing insights into potential resistance mechanisms present in the genomes.

## Secondary metabolites analysis in *P. acidilactici*

The identification and analysis of secondary metabolites in the Thai *P. acidilactici* strains utilized AntiSMASH v6.0, allowing comprehensive detection of biosynthetic gene clusters [42], and BAGEL4 for its targeted prediction of bacteriocin genes [43]. This combination of tools provided a dual approach to explore not only the broad spectrum of potential secondary metabolites but also specific antimicrobial compounds critical for probiotic efficacy.

## Genome stability assessment in *P. acidilactici*

The stability of the genomes of the five Thai *P. acidilactici* strains was evaluated using several bioinformatics tools. CRISPRCasFinder v1.1.2 was employed to predict CRISPR and cas genes associated with bacterial immunity [44], while PHASTER was utilized to identify prophage sequences [45], PlasmidFinder to detect plasmids ([46]), OriTFinder to search for origin of transfer (OriT) regions (https://bioinfo-mml.sjtu.edu.cn/oriTfinder/) [47], and IslandViewer 4 (incorporating SIGI-HMM, IslandPath-DIMOB, and IslandPick algorithms) to identify genomic islands [48]. This comprehensive approach provided insights into the genetic stability and potential for horizontal gene transfer (HGT) in the Thai LAB strains.

## Analysis of probiotic functional genes and metabolic pathways in *P. acidilactici*

Probiotic functional genes and metabolic pathways in Thai *P. acidilactici* strains were meticulously analyzed by mining annotation files for relevant keywords and investigating functional subsystems via the RAST (Rapid Annotation using Subsystem Technology) server, which facilitates a nuanced understanding of bacterial functionalities. BLASTp comparisons with reference protein sequences further refined the specificity of gene identification. Metabolic pathways were then

elucidated using the KEGG (Kyoto Encyclopedia of Genes and Genomes) Mapper Reconstruct Pathway tool, with Blast-KOALA assigning KEGG identifiers (K numbers) to ensure accurate pathway mapping [49,50].

## Results

### Species confirmation of *P. acidilactici* strains

Species confirmation of *P. acidilactici* strains was achieved using MALDI-TOF analysis, with all five Thai LAB strains scoring between 2.10 and 2.36, indicating reliable species-level identification (S2 Table). In contrast, the VITEK 2 Compact system demonstrated lower discriminatory power, although it accurately identified strain AF2019 as *P. acidilactici* with a high confidence score of 94%.

### Antimicrobial susceptibility testing of *P. acidilactici* strains

Antimicrobial susceptibility testing of the five Thai *P. acidilactici* strains was conducted according to EFSA guidelines. The MICs obtained indicated that strains IAF6519, IAF5919, and P72N were susceptible to all tested antibiotics, except for vancomycin, to which all strains exhibited intrinsic resistance. In contrast, strains AF2519 and AF2019 showed resistance to erythromycin and clindamycin, with AF2519 additionally being resistant to tetracycline. The detailed MIC results, categorized by EFSA's predefined cut-off values, are presented in Table 1.

### Hemolytic activity of *P. acidilactici* strains

The hemolytic activity of the five Thai *P. acidilactici* strains was assessed to gain more insight about their potential safety as probiotics. All strains exhibited alpha-hemolysis, which involves partial disruption of red blood cells without complete lysis, as depicted in S1 Fig.

### Genome properties of *P. acidilactici* strains

WGS of the five Thai *P. acidilactici* strains revealed genome sizes ranging from 1.81 to 2.19 Mb, with a consistent G + C content between 41.9% and 42.2%, as shown in Table 2. The genomic assemblies achieved a completion rate of 99.4% with no detectable contamination, underscoring the reliability and accuracy of our sequencing results. The number of coding sequences (CDS) varied across the strains, reflecting diversity in their genomic content. Notably, strains IAF5919, AF2519 and AF2019 harbored plasmids, which are known to potentially enhance probiotic capabilities. However, the

**Table 1. Antibiotic susceptibility profiles of five Thai *P. acidilactici* strains.**

| Antibiotics | Cut-off (mg/L) | IAF6519 | IAF5919 | AF2519 | AF2019 | P72N |
|---|---|---|---|---|---|---|
| Ampicillin | 4 | 2 | 2 | 1 | 1 | 0.5 |
| Clindamycin | 1 | ≤ 0.25 | ≤0.25 | 4 | 4 | ≤0.25 |
| Gentamicin | 16 | ≤1 | ≤1 | ≤1 | ≤1 | ≤1 |
| Erythromycin | 1 | ≤0.25 | ≤0.25 | 4 | 4 | ≤0.25 |
| Kanamycin | 64 | ≤16 | ≤16 | ≤16 | ≤16 | ≤16 |
| Tetracycline | 8 | 8 | 8 | 16 | 8 | 4 |
| Streptomycin 1,000 ug/ml | 64 | ≤1,000 | ≤1,000 | ≤1,000 | ≤1,000 | ≤1,000 |
| Chloramphenicol | 4 | ≤8 | ≤8 | ≤8 | ≤8 | ≤8 |
| Vancomycin | N. R | >32 | >32 | >32 | >32 | >32 |

Yellow-shaded boxes show MIC of antibiotics higher than the corresponding cut-off values used by the European Food Safety Authority [16]. N.R: not required.

**Table 2. Genome properties of five Thai *P. acidilactici* strains.**

| Strain | No. of contigs | Total length (bp) | Largest contigs (bp) | Contig N50 (bp) | Contig L50 | G + C content (%) | Completeness | Contamination | Plasmid |
|--------|----------------|-------------------|----------------------|-----------------|------------|-------------------|--------------|---------------|---------|
| IAF6519 | 1 | 1,984,484 | 1,984,484 | 1,984,484 | 1 | 42.2 | 99.4 | 0 | Not detected |
| IAF5919 | 2 | 2,021,060 | 1,984,504 | 1,984,504 | 1 | 42.2 | 99.4 | 0 | Detected |
| IAF2519 | 4 | 2,228,155 | 2,228,155 | 2,228,155 | 1 | 41.9 | 99.4 | 0 | Detected |
| IAF2019 | 4 | 2,220,091 | 2,138,608 | 2,138,608 | 1 | 42.0 | 99.4 | 0 | Detected |
| P72N | 1 | 1,953,388 | 1,953,388 | 1,953,388 | 1 | 42.2 | 99.4 | 0 | Not detected |

presence of plasmids in these strains also introduces the possibility of HGT. In contrast, strains IAF6519 and P72N were found to lack plasmids. Detailed circular genome maps, illustrating key genomic features, are provided in Fig 1.

### Strain identification and phylogenetic tree analysis of *P. acidilactici* strains

Species-level identification based on OrthoANI analysis confirmed that all five Thai isolates belong to *P. acidilactici*, showing >97% pairwise identity with the reference strain PMC65 (Fig 2). The high genomic similarity among strains, particularly between IAF6519 and IAF5919—was further supported by core SNP-based maximum likelihood phylogeny (Fig 3), reflecting close evolutionary relationships consistent with their shared origin from chicken fecal samples. A broader

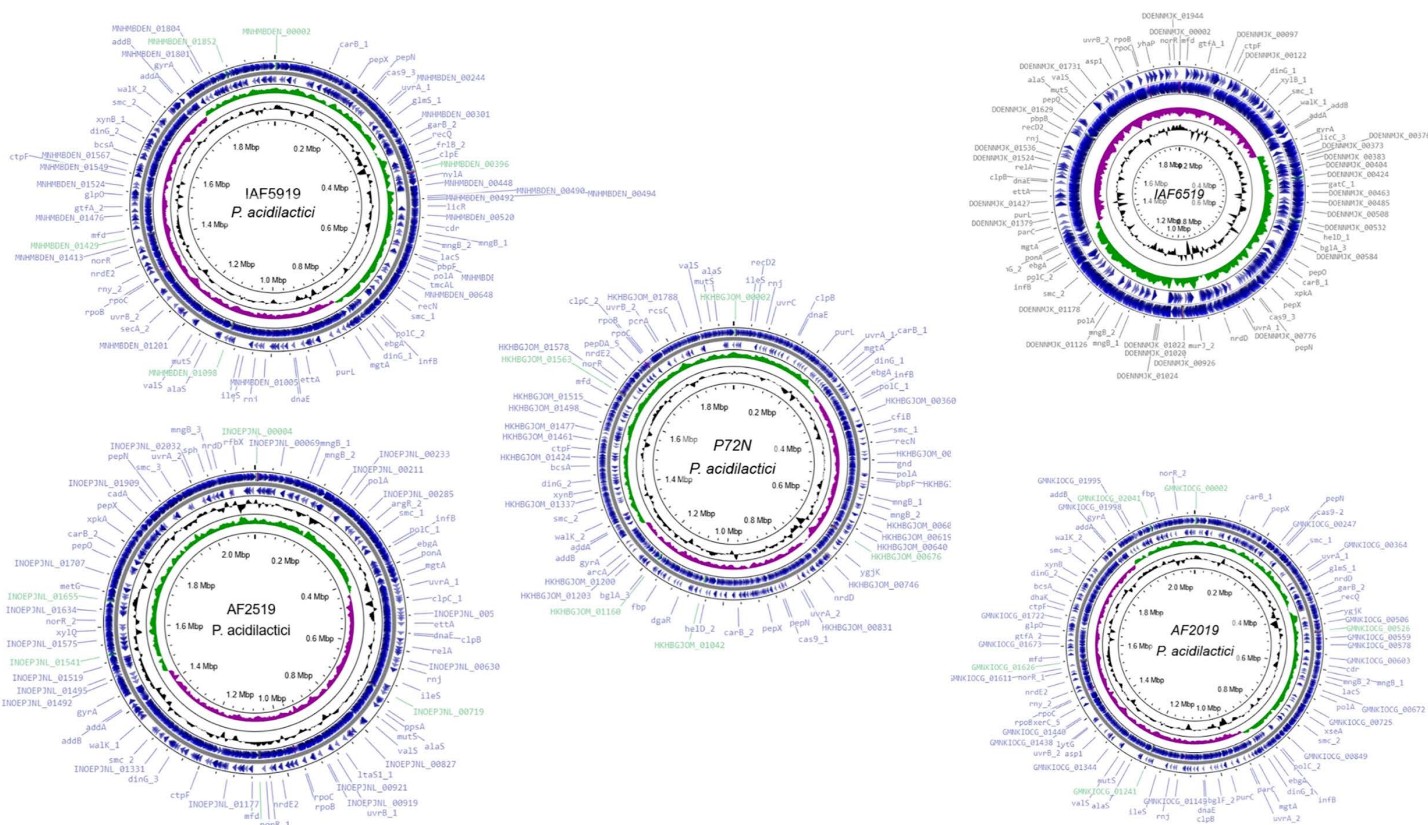

**Fig 1. Circular genome maps of five Thai *P. acidilactici* strains.** The first circle presents CRISPR repeats in black. The second circle depicts the GC skew (G + C/G − C), with values >0 in green and values <0 in purple. The third circle denotes rRNA, tRNA, and the sites of CDS.

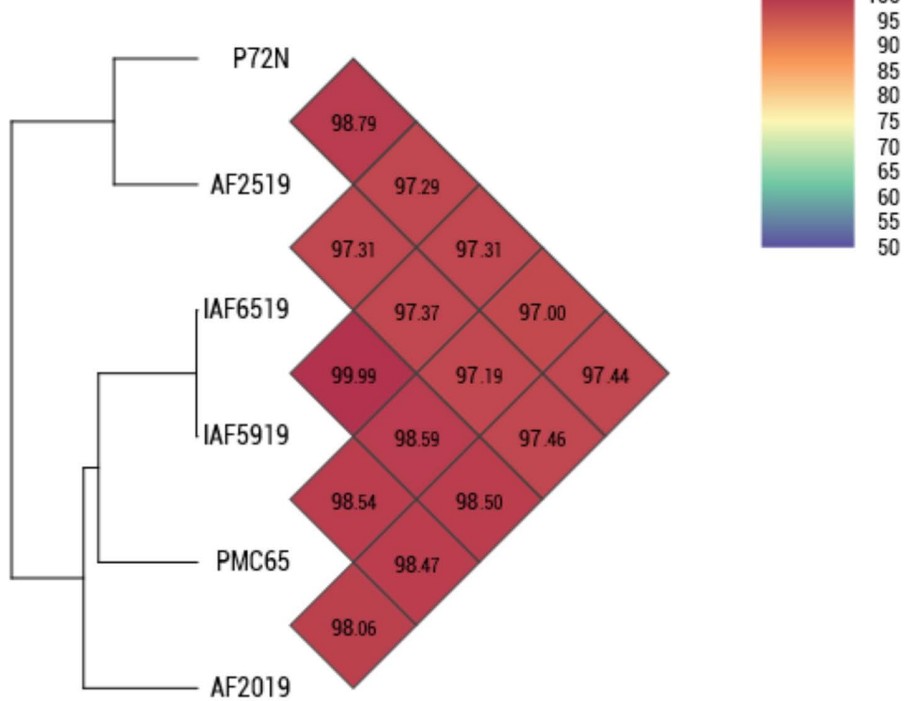

**Fig 2. OrthoANI species confirmation of five *P. acidilactici* strains compared to the reference strain *P. acidilactici* PMC65.**

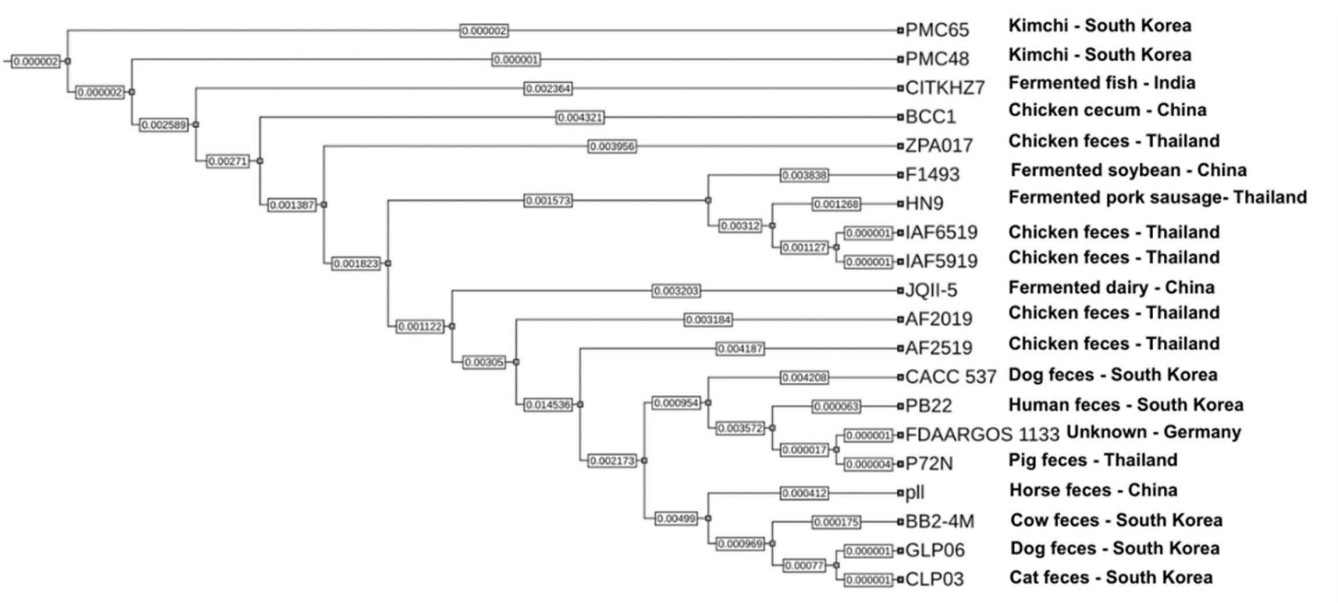

**Fig 3. Phylogenetic tree of *P. acidilactici* strains based on core SNPs.**

comparison including additional *P. acidilactici* and related species is provided in S2 Fig. A total of 2,715 orthologous clusters were identified across six *P. acidilactici* genomes, including five Thai isolates and the reference strain HN9 (Fig 4). Of these, 1,546 clusters were conserved across all strains. The number of strain-specific orthologous clusters varied among the genomes. HN9 contained 23 strain-specific clusters, followed by AF2019 with 11, AF2519 with 9, and P72N with 1. No

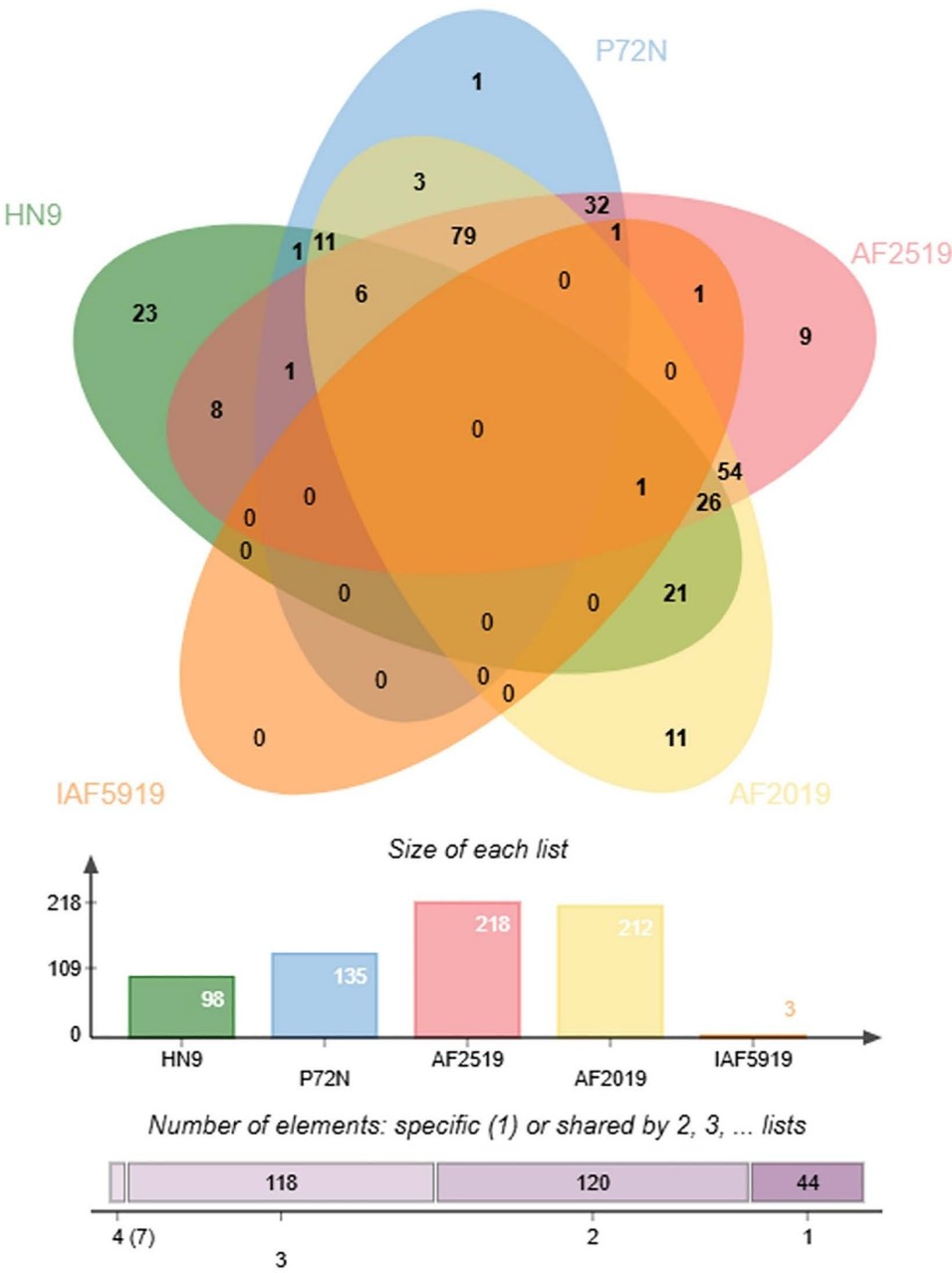

**Fig 4. Orthologous gene cluster distribution among Thai *P. acidilactici* isolates and *P. acidilactici* HN9 The bars show the number of clusters for each strain.** The purple boxes display the values of shared or single orthologous clusters.

strain-specific clusters were detected in IAF5919 or IAF6519. A complete list of strain-specific clusters and their corresponding gene annotations is provided in S3 Table.

### Identification of antimicrobial resistance genes of *P. acidilactici* strains

WGS revealed the presence and distribution of AMR genes across the five Thai *P. acidilactici* strains. The CARD analysis indicated that strains IAF6519, IAF5919 and P72N lacked notable AMR genes, suggesting a low risk profile for these strains, as shown in Table 3. In contrast, ResFinder results showed strain AF2519 harbored *erm(B)* and *lnu(A)* on unclassified DNA elements, with *tet(M)* located on plasmid pAF25191. Strain AF2019 exhibited a broader array of AMR genes, including *erm(B)*, *NarA*, and *NarB* on plasmid pAF20193, along with *tet(M)* on plasmid pAF20191 (S4 Table). Additional KEGG analysis identified chromosome-associated AMR-related genes such as *dltA*, *dltB*, *dltC*, *dltD*, *penP*, and *lsa* in strains IAF6519 and IAF5919, which may impact their overall resistance profiles. Strains AF2519 and AF2019 displayed a more diverse range of resistance genes across multiple plasmids, suggesting enhanced adaptability and resistance capabilities. Detailed gene locations and summaries are provided in S3 Fig.

**Table 3. Antimicrobial resistance genes identified in the genomes of five *P. acidilactici* strains (CARD analysis).**

| Strain | Gene | Molecular Type | AMR Gene Family | Resistance Mechanism | % Identity of Matching Region |
|---|---|---|---|---|---|
| IAF6519 | *vanT* in *vanG* cluster | Chromosome | glycopeptide resistance gene cluster, *vanT* | antibiotic target alteration | 31.15 |
| | *qacG* | Chromosome | small multidrug resistance (SMR) antibiotic efflux pump | antibiotic efflux | 48.11 |
| | *sdrM* | Chromosome | major facilitator superfamily (MFS) antibiotic efflux pump | antibiotic efflux | 34.81 |
| IAF5919 | *vanT* in *vanG* cluster | Chromosome | glycopeptide resistance gene cluster, *vanT* | antibiotic target alteration | 31.15 |
| | *sdrM* | Chromosome | major facilitator superfamily (MFS) antibiotic efflux pump | antibiotic efflux | 34.81 |
| | *qacG* | Chromosome | small multidrug resistance (SMR) antibiotic efflux pump | antibiotic efflux | 48.11 |
| AF2519 | *vanT* in *vanG* cluster | Chromosome | glycopeptide resistance gene cluster, *vanT* | antibiotic target alteration | 31.69 |
| | *qacG* | Chromosome | small multidrug resistance (SMR) antibiotic efflux pump | antibiotic efflux | 48.11 |
| | *tet(M)* | Plasmid | tetracycline-resistant ribosomal protection protein | antibiotic target protection | 99.37 |
| | *ermB* | DNA element | Erm 23S ribosomal RNA methyltransferase | antibiotic target alteration | 98.79 |
| | *lnuA* | DNA element | lincosamide nucleotidyltransferase (LNU) | antibiotic inactivation | 97.52 |
| AF2019 | *sdrM* | Chromosome | major facilitator superfamily (MFS) antibiotic efflux pump | antibiotic efflux | 34.81 |
| | *vanT* in *vanG* cluster | Chromosome | glycopeptide resistance gene cluster, *vanT* | antibiotic target alteration | 31.42 |
| | qacG | Chromosome | small multidrug resistance (SMR) antibiotic efflux pump | antibiotic efflux | 48.11 |
| | *tet(M)* | Plasmid | tetracycline-resistant ribosomal protection protein | antibiotic target protection | 99.37 |
| | *ermB* | Plasmid | Erm 23S ribosomal RNA methyltransferase | antibiotic target alteration | 99.18 |
| P72N | *sdrM* | Chromosome | major facilitator superfamily (MFS) antibiotic efflux pump | antibiotic efflux | 34.81 |
| | *qacG* | Chromosome | small multidrug resistance (SMR) antibiotic efflux pump | antibiotic efflux | 48.11 |
| | *vanT* in *vanG* cluster | Chromosome | glycopeptide resistance gene cluster, *vanT* | antibiotic target alteration | 31.42 |

### Identification of genes encoding virulence factors in the *P. acidilactici* strains

Genomic analysis of the five Thai *P. acidilactici* strains revealed the presence of several genes associated with core cellular functions and virulence, as detailed in Fig 5. Common genes across all strains included endopeptidase Clp ATP-binding chain C (*clpC*), enolase, elongation factor, glyceraldehyde-3-phosphate dehydrogenase, and chaperonin *GroEL*. Notably, the ATP-dependent Clp protease proteolytic subunit gene (*clpP*) was absent in strain AF2519, while the gene encoding dTDP-glucose-4,6-dehydratase (*rmlB*), involved in dTDP-rhamnose biosynthesis, was uniquely present in AF2019. Major virulence genes such as gelatinase (*gelE*), hyaluronidase (*hyl*), and enterococcal surface protein (*esp*) were absent in all strains. Additionally, the Enterolysin A biosynthetic gene cluster, linked to bacteriocin production and potential antimicrobial properties, was detected in all strains except P72N (S5 Table).

### Genome stability assessment of *P. acidilactici* strains

**CRISPR-Cas systems in *P. acidilactici* strains.** All five Thai strains were found to contain a Type IIA CRISPR-Cas system, as identified using CRISPRCasFinder v1.1.2 (S6 Table). In strain IAF6519, the cas9_TypeII gene was located on the forward strand, and in strain P72N, the cas9_TypeII and csn2_TypeIIA genes were located on the reverse strand.

**Prophage sequences of *P. acidilactici* strains.** Prophage regions were profiled using PHASTER. Strains IAF6519 and IAF5919 each contained one intact chromosomal prophage region (S7 Table). Strain AF2519 had two intact and three

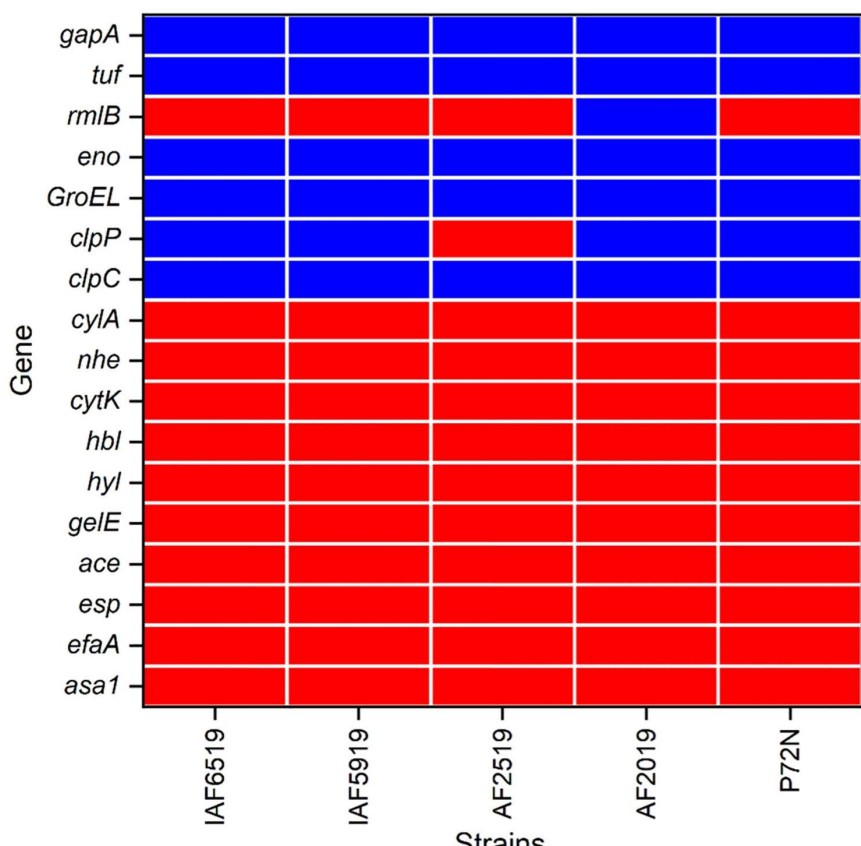

**Fig 5. Presence of known virulence factor genes in five Thai *P. acidilactici* strains.** Blue indicates the presence of the virulence factor genes, while red indicates their absence.

incomplete prophage regions, plus two plasmid regions. Strain AF2019 contained four intact chromosomal prophage regions and one plasmid region of questionable completeness. Strain P72N had one intact chromosomal prophage region.

**Insertion sequences and composite transposons of *P. acidilactici* strains.** Insertion sequences and composite transposons were identified in all strains (S8 Table). IAF6519 contained two ISLpl1 insertion sequences and one cn_4153_ISLpl1 composite transposon on the chromosome. IAF5919 had two ISLpl1 sequences and one cn_4153_ISLpl1 composite transposon on the chromosome, plus one ISLhe30 and one ISS1N on a plasmid. AF2519 featured one ISLhe30 and one ISLpl1 on a plasmid, plus one ISS1N and two composite transposons (cn_18158_ISLpl1 and cn_10454_ISS1N). AF2019 exhibited two ISLpl1 sequences and one cn_7438_ISLpl1 composite transposon on the chromosome, plus one ISLhe30 and two ISLpl1 sequences on plasmids. P72N had two ISLpl1 insertion sequences and one cn_4466_ISLpl1 composite transposon on the chromosome.

## Plasmid and genomic island analysis

Genomic analysis revealed multiple plasmids within the *P. acidilactici* strains, each carrying diverse CDS. Notably, strain AF2519 harbored three distinct plasmids: plasmid pAF25191, which included genes associated with nutrient transport and stress resistance, such as *tet(M)*, *oppB*, and *dppE*; plasmid pAF25192, carrying genes potentially involved in secondary metabolite production and antibiotic resistance, including *cypC* and *mccF*; and an unclassified DNA element containing resistance genes such as *erm(B)*. The plasmid in strain IAF5919, pIAF5919, contained genes linked to carbohydrate metabolism and stress adaptation, as detailed in Fig 6. OriTfinder analysis indicated that all strains possessed a single chromosomal relaxase gene but lacked crucial conjugation elements (T4CP and oriT) on the plasmids of strains AF2519, AF2019, and IAF5919, suggesting a limited potential for HGT (S9 Table). Additionally, IslandViewer4 analysis identified four genomic islands in strain AF2019, all located on the plasmid pAF20191. These islands feature genes such as *tet(M)* and insertion sequences, indicating genomic plasticity (S10 Table).

## Identification and analysis of probiotic functional genes and metabolic pathways

RAST (Rapid Annotation using Subsystem Technology) analysis of the five Thai *P. acidilactici* strains revealed diverse gene distributions across various subsystems. Notably, the 'Carbohydrates' subsystem was the most prominent, with strains AF2019 and AF2519 exhibiting the highest gene counts, suggesting robust capabilities in carbohydrate metabolism, as detailed in S11 Table. These strains also showed elevated gene counts in the 'Phages, Prophages, Transposable Elements, Plasmids' subsystem, indicative of significant genomic plasticity.

KEGG orthology analysis identified strain-specific variations in metabolic pathways. In the 'Carbohydrate Metabolism' category, strain AF2519 displayed the highest gene count. Additionally, the 'Protein Families: Genetic Information Processing' category revealed notable variations among the strains, with AF2519 again leading in gene counts, as shown in S12 Table. This comprehensive gene profiling highlighted the metabolic diversity and potential adaptability of these strains to various environmental conditions.

## Probiotic-related genes of *P. acidilactici* strains

Genomic analysis identified probiotic-related genes grouped into seven functional categories: GIT survival and stress response, heat stress, acid stress, bile tolerance, adhesion, antioxidant defense, and immunomodulation (Table 4). All five Thai *P. acidilactici* strains were found to possess representative genes in all seven categories. These include *ponA*, *pbpX* (GIT survival); *grpE*, *groS* (heat stress); *atpE*, *nhaC* (acid stress); *ppaC*, *cfa* (bile tolerance); *tuf*, *eno* (adhesion); *katA*, *trxA* (antioxidant defense); and *dltA*, *dltC* (immunomodulation). A comprehensive list of probiotic-related genes identified across all strains is provided in S13 Table

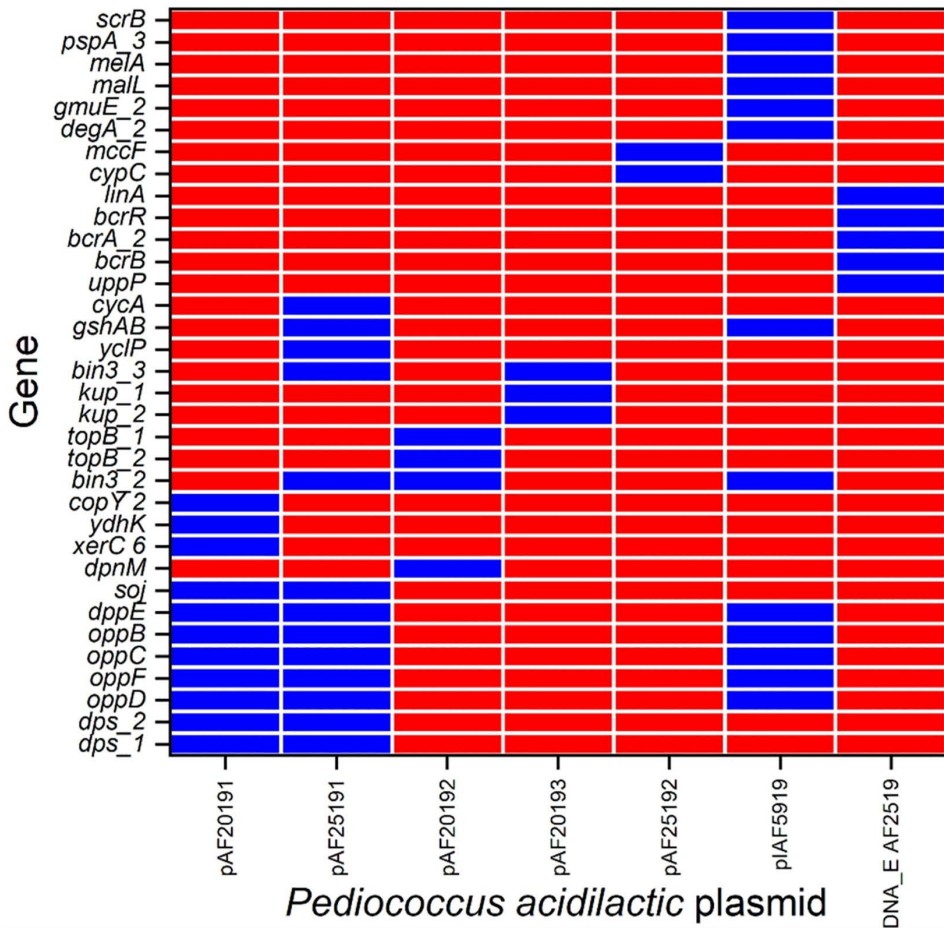

**Fig 6. Genes identified in the plasmids of the five Thai _P. acidilactici_ strains by using the KEGG database.** Blue indicates the presence of the virulence factor genes, while red indicates their absence.

**Table 4. Representative probiotic-related genes across seven functional categories in five _P. acidilactici_ strains.**

| Functional Category | Genes Representative | IAF6519 | IAF5919 | AF2519 | AF2019 | P72N |
|---|---|---|---|---|---|---|
| GIT survival and stress response | _ponA, pbpX_ | + | + | + | + | + |
| Heat stress | _grpE, groS_ | + | + | + | + | + |
| Acid stress | _atpF, nhaC_ | + | + | + | + | + |
| Bile tolerance | _ppaC, cfa_ | + | + | + | + | + |
| Adhesion | _tuf, eno_ | + | + | + | + | + |
| Antioxidant defense | _katA, trxA_ | + | + | + | + | + |
| Immunomodulation | _dltA, dltC_ | + | + | + | + | + |

## Discussion

Probiotics are increasingly explored as safe alternatives to antibiotics for some applications in livestock, but comprehensive safety assessments remain critical, particularly regarding AMR. In this study, WGS and functional profiling were used

to assess five *P. acidilactici* strains isolated from poultry and swine, focusing on both probiotic attributes and genomic safety indicators.

Phenotypic antimicrobial susceptibility testing (AST) revealed that most strains were susceptible to the antibiotics tested, except for consistent resistance to vancomycin across all isolates. This pattern is consistent with previous findings that vancomycin resistance in LAB, including *P. acidilactici*, is an intrinsic trait due to the presence of D-Ala-D-Lac termini in peptidoglycan precursors [51]. In contrast, two strains (AF2519 and AF2019) exhibited elevated MIC against erythromycin, clindamycin, and tetracycline, suggesting possible acquired resistance.

WGS of *P. acidilactici* AF2519 and AF2019 revealed that both strains harbor plasmids encoding classical AMR genes specifically *tet(M)*, *erm(B)*, and, in the case of AF2519, *lnu(A)*. These findings are striking because most previously characterized *P. acidilactici* strains have shown either intrinsic or chromosomal ARGs, with plasmid-mediated resistance rarely reported [52,53]. In contrast, our isolates possess composite resistance loci located on plasmids, suggesting recent acquisition and higher mobility potential. This challenges the prevailing view of *P. acidilactici* as genetically stable and free of transferable ARGs.

Although the AMRG-carrying plasmids lacked canonical conjugation machinery (e.g., *oriT*, T4CP, T4SS), in trans mobilization cannot be ruled out. Interestingly, AF2019 harbored an additional plasmid (pAF20192) that contains a complete conjugation machinery, including relaxase, T4CP, and T4SS components, yet notably lacked any ARGs. This configuration suggests that pAF20192 could serve as a helper plasmid, facilitating mobilization of resistance-carrying elements via trans mechanisms under gut-like conditions [54]. Similar instances of *tet(M)* and *erm(B)* transfer from *Lactobacillus* to *Enterococcus* under gut conditions previously have been documented [55].

Furthermore, AF2019 was the only strain found to contain four genomic islands, all located on plasmid pAF20191, one of which co-localized *tet(M)* with multiple insertion sequences. Genomic islands are recognized hotspots for horizontal gene acquisition and dissemination, especially when harboring ARGs flanked by mobile elements. The fact that these islands reside on a plasmid, an inherently mobile genetic element, further increases the risk of horizontal transfer. This genomic configuration mirrors composite transposons described in *E. coli* and *Streptococcus pyogenes* [56,57], suggesting that these loci may represent evolutionary remnants of past mobilization events and retain the potential for future dissemination.

From a regulatory perspective, plasmid localization of *tet(M)* and *erm(B)* elevates concern. While chromosomal *tet(M)* has been reported in strain FAM 18,375 [58], the plasmid-borne configuration in our strains implies greater potential for dissemination. In line with EFSA guidance, strains harboring transferable ARGs, regardless of phenotypic resistance, should be excluded from probiotic use [59]. These findings reinforce the importance of incorporating mobility context into AMR screening protocols for probiotic candidates.

AF2019 exhibited notable genotype–phenotype discrepancies: these included susceptibility to tetracycline despite the presence of plasmid-borne *tet(M)*, and resistance to clindamycin without detectable resistance genes. Such mismatches, reported in LAB and other Gram-positive bacteria, may result from several molecular mechanisms [60–62]. For tetracycline susceptibility, possible explanations include: (i) promoter inactivity or mutations in upstream regulatory regions reducing transcription efficiency [63,64], (ii) structural mutations in the *tet(M)* coding sequence (e.g., frameshift, missense, or nonsense mutations) impairing ribosomal protection [63], and (iii) inducible expression or gene silencing under laboratory conditions lacking environmental cues necessary for activation [64]. For clindamycin resistance without *erm* or *lnu* genes, potential mechanisms include: (i) chromosomal point mutations in 23S rRNA domain V or ribosomal proteins L4/L22 altering antibiotic binding [65], (ii) overexpression of multidrug efflux pumps such as MFS transporters capable of extruding lincosamides [66], and (iii) novel or uncharacterized resistance determinants not represented in current databases, such as the *cfr* gene, which methylates A2503 of 23S rRNA and confers cross-resistance to lincosamides and other antibiotic classes [65]. These possibilities underscore that phenotypic assays alone may underestimate hidden AMR risks, while purely genomic screens may overlook cryptic or novel resistance mechanisms, reinforcing the need for integrated phenotypic, genomic, and transcriptomic approaches in probiotic safety evaluation.

Beyond identifying individual AMR genes, the structural organization of resistance loci in AF2519 and AF2019 provides insight into their evolutionary and mobilization potential. In contrast to previous reports of *tet(M)* in chromosomal or isolated contexts [67], our strains harbor a plasmid-borne cluster of *tet(M)*, *erm(B)*, and *lnu(A)* co-localized with insertion sequences and recombinases, suggesting a mosaic scaffold shaped by historical recombination. Even without a complete conjugation system, such architecture may retain latent mobility under ecological pressure. Collectively, these findings challenge the assumption that resistance genes in probiotics are static or benign and highlight the value of high-resolution structural analysis in probiotic risk evaluation.

The genomes of the five Thai strains encoded a comprehensive set of stress-tolerance and host-interaction functions. These included high–molecular–weight penicillin-binding proteins (PBP1A/*ponA*, PBP2X/*pbpX*) that support peptidoglycan synthesis and cell-wall integrity under gut stress [68]. Classic heat-shock chaperones such as *grpE* and *groS* also were identified, consistent with LAB where *GroESL* systems enhance thermal survival during feed processing [69]. Acid tolerance genes, including *atpF* and *nhaC*, mediate pH homeostasis via proton extrusion [70]. For bile adaptation, *ppaC* and *cfa* were present; notably *cfa* (cyclopropane fatty acid synthase) modifies membrane lipids and has been linked to stress resilience in *Lactococcus* and *Bacillus* [71].

Surface-adhesion factors *tuf* (EF-Tu) and *eno* (enolase) were also detected, consistent with their roles as "moonlighting" adhesins in mucosal binding [72,73]. Oxidative stress defenses include *trxA* and *katA*; while *trxA* is ubiquitous in LAB [74], *katA* is rare in *Pediococcus* and suggests acquisition of catalase activity [75]. The presence of a complete *dlt* operon (*dltA*, *dltC*) indicates D-alanylation of teichoic acids, a modification known to enhance resistance to host antimicrobial peptides [76]. Notably, *katA* (heme-dependent catalase) and *cfa* (cyclopropane fatty acid synthase) have not been previously reported in *P. acidilactici*. The rarity of *katA* in lactic acid bacteria is linked to their fermentative metabolism and lack of a respiratory chain, suggesting its presence in our strains may have resulted from horizontal acquisition from catalase-positive bacteria in the gut or feed environment [77,78]. Catalase activity decomposes hydrogen peroxide, enhancing oxidative stress tolerance, improving survival during aerobic storage, and supporting viability in the gastrointestinal tract [79,80]. The *cfa* gene modifies membrane phospholipids to produce cyclopropane fatty acids, increasing membrane rigidity and hydrophobicity, which improves acid and bile tolerance, stability during feed processing, and persistence in intestinal environments [81,82]. The co-occurrence of *katA* and *cfa* in these strains may therefore represent a unique adaptive genomic signature, enhancing resilience during gut transit and feed storage, traits that directly contribute to their suitability as probiotic candidates for use in poultry and swine production [83].

The findings of this study present a refined view of *P. acidilactici* strains as probiotic candidates. While two strains carried plasmid-borne resistance genes, all lacked virulence markers and retained key probiotic traits. Although all five strains exhibited alpha-hemolysis on blood agar, this is a common and non-pathogenic characteristic among lactic acid bacteria, including *P. acidilactici*, and does not imply virulence [84]. Alpha-hemolysis, characterized by partial lysis of red blood cells, is frequently observed in probiotic LAB such as *Lacticaseibacillus rhamnosus* and *Lactococcus lactis*, and is generally considered safe in food and probiotic applications [85,86]. These findings underscore the need for strain-level evaluation grounded in genomic context rather than binary classification. The presence of rare genes such as *katA* and *cfa* further suggests host-adaptive potential. As probiotics gain prominence in livestock systems, whole-genome analysis should serve as a cornerstone for precision selection and responsible application.

## Conclusions

This study demonstrates how whole-genome sequencing, when paired with targeted functional analysis, provides a high-resolution view of probiotic strain safety and performance [60,62,50,61]. Among the five *P. acidilactici* strains analyzed, three exhibited uncomplicated genomic backgrounds with strong probiotic features, while two harbored plasmid-borne ARGs with limited mobilization capacity. Whole-genome data also uncovered rare but functionally relevant genes, such as *katA* and *cfa* that may enhance oxidative and bile stress resilience, respectively. These findings reinforce

the limitations of phenotypic-only approaches and affirm the importance of genomic context in evaluating both safety and functionality. As probiotics gain increasing relevance in livestock systems, this study positions WGS as a foundational tool for evidence-based strain selection, ecological fit, and regulatory assurance.

## Supporting information

**S1 Table. *P. acidilactici* strains used for phylogenetic analysis.** * The genomic sequences of the 15 control strains were obtained from the NCBI database. The accession numbers correspond to the genomes of the control strains.
(XLSX)

**S2 Table. MALDI-TOF and VITEK confirmation of *P. acidilactici* species identity.**
(XLSX)

**S1 Fig. Hemolytic Activity Test of five Thai *P. acidilactici* strains.** 1: IAF5919; 2: IAF6519; 3: AF2519; 4: AF2019; 5: P72N; 6: *Bacillus sp.*
(TIF)

**S3 Table. Unique clusters in strains AF2019, AF2519, and P72N from OrthoVenn3 analysis with HN9 as a reference.**
(XLSX)

**S4 Table. Antimicrobial resistance genes identified in the genome sequences of the five Thai.** *P. acidilactici* strains by using ResFinder.
(XLSX)

**S2 Fig. OrthoANI heatmap showing pairwise average nucleotide identity (ANI) values among the five Thai *P. acidilactici* strains and additional reference strains from diverse *Pediococcus* species.** All five Thai strains shared >97% ANI with the reference *P. acidilactici* strain PMC65, confirming species identity. In contrast, ANI values between these and strains such as *WCF51*, *L25F*, and *AF2319* were substantially lower (<70%), supporting their taxonomic distinction and reaffirming the correct species-level classification.
(TIF)

**S3 Fig. Antimicrobial resistance genes identified in the genome sequences of the five Thai *P. acidilactici* strains by using the KEGG database.** Blue indicates the presence of the virulence factor genes, while red indicates their absence.
(TIF)

**S5 Table. Identification and analysis of the Enterolysin A gene in the five Thai *P. acidilactici.* Strains.**
(XLSX)

**S6 Table. CRISPR-Cas loci identified in the five Thai *P. acidilactici* strains.**
(XLSX)

**S7 Table. Prophage regions predicted in the five Thai *P. acidilactici* strains.**
(XLSX)

**S8 Table. Insertion sequences and composite transposons in the five Thai *P. acidilactici* strains.**
(XLSX)

**S9 Table. Results of the OriTfinder analysis of plasmids in the five Thai *P. acidilactici* strains.**
(XLSX)

**S10 Table. Predicted genomic islands, their sizes, and the methods used to identify them on plasmid pAF20191 of *Pediococcus acidilactici* strain AF2019.**
(XLSX)

**S11 Table. RAST annotation of functional subsystems in the five Thai *P. acidilactici* strains.**
(XLSX)

**S12 Table. KEGG pathway classification of annotated genes in the five Thai *P. acidilactici.* Strains.**
(XLSX)

**S13 Table. Full probiotic-related gene list across 12 categories in the five Thai *P. acidilactici.* Strains.**
(XLSX)

## Acknowledgments

We extend our gratitude to the staff of the Department of Veterinary Microbiology, Faculty of Veterinary Science, Chulalongkorn University for their technical support. We also thank the Ratchadaphisek Somphot Scholarship, Chulalongkorn University, for supporting the presentation of our work at an international academic conference.

## Author contributions

**Conceptualization:** Nuvee Prapasarakul.

**Data curation:** Yuda Disastra, Thidathip Wongsurawat, Piroon Jenjaroenpun, David John Hampson.

**Formal analysis:** Yuda Disastra, Thidathip Wongsurawat, Piroon Jenjaroenpun, Ratchnida Kamwa.

**Funding acquisition:** Nuvee Prapasarakul.

**Investigation:** Yuda Disastra, Ratchnida Kamwa.

**Methodology:** Yuda Disastra, Thidathip Wongsurawat, Piroon Jenjaroenpun, Ratchnida Kamwa.

**Project administration:** Nuvee Prapasarakul.

**Resources:** Yuda Disastra, Thidathip Wongsurawat, Piroon Jenjaroenpun, Nuvee Prapasarakul.

**Supervision:** David John Hampson, Nuvee Prapasarakul.

**Validation:** Yuda Disastra, Piroon Jenjaroenpun, David John Hampson, Nuvee Prapasarakul.

**Visualization:** Yuda Disastra, Thidathip Wongsurawat, Ratchnida Kamwa.

**Writing – original draft:** Yuda Disastra, Nuvee Prapasarakul.

**Writing – review & editing:** David John Hampson, Nuvee Prapasarakul.

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
