## [Decision Letter · Decision Letter 0]

5 Aug 2025

PONE-D-25-35195Integrative genomic characterization of Pediococcus acidilactici strains reveals differing probiotic safety profilesPLOS ONE

Dear Dr. Prapasarakul,

Thank you for submitting your manuscript to PLOS ONE. After careful consideration, we feel that it has merit but does not fully meet PLOS ONE’s publication criteria as it currently stands. Therefore, we invite you to submit a revised version of the manuscript that addresses the points raised during the review process. 

We look forward to receiving your revised manuscript.

Kind regards,

Míriam R. García

Academic Editor

PLOS ONE

2. Please note that your Data Availability Statement is currently missing [the repository name and/or the DOI/accession number of each dataset OR a direct link to access each database]. If your manuscript is accepted for publication, you will be asked to provide these details on a very short timeline. We therefore suggest that you provide this information now, though we will not hold up the peer review process if you are unable.

Additional Editor Comments (if provided):

Reviewers' comments:

Reviewer's Responses to Questions

**Comments to the Author**

1. Is the manuscript technically sound, and do the data support the conclusions?

Reviewer #1: Yes

2. Has the statistical analysis been performed appropriately and rigorously? 

Reviewer #1: N/A

3. Have the authors made all data underlying the findings in their manuscript fully available?

Reviewer #1: Yes

4. Is the manuscript presented in an intelligible fashion and written in standard English?

Reviewer #1: Yes

5. Review Comments to the Author

Reviewer #1: Overall comment:

1. The manuscript is well-organized and written.

2. The discussion is well-reasoned and integrates genetic discoveries with regulatory concerns, including the EFSA recommendations.

Recommendations

1. Abstract: "novel composite alignment" of tet(M); this terminology could be explained more precisely to aid the general readership.

2. Discussion: The observed susceptibility to tetracycline in the presence of tet(M) and resistance to clindamycin in the absence of comparable ARGs warrants further mechanistic discussion. Could gene silence, promoter inactivity, or undiscovered resistance factors be involved? Please discuss more about it.

3. Discussion: The discovery of katA and cfa, which are rarely seen in P. acidilactici, is interesting. These findings indicate potential innovative adaptations. Please find literature or brief contextualization to discuss.

4. S1 Fig. Hemolytic Activity Test of five Thai P. acidilactici strains: If you have a clearer image, please replace it for plate marked no. 5 and 6

6. PLOS authors have the option to publish the peer review history of their article (what does this mean? ). If published, this will include your full peer review and any attached files.

**Do you want your identity to be public for this peer review?** For information about this choice, including consent withdrawal, please see our Privacy Policy .

Reviewer #1: No

---

## [Author Response · Author response to Decision Letter 1]

26 Aug 2025

Dear Editor

We thank the reviewer for the positive assessment of our manuscript and the constructive comments that have helped us to improve the clarity and contextualization of our findings. We have addressed each point as detailed below, with changes highlighted in the “Revised Manuscript with Track Changes” file.

Comment 1: Abstract: "novel composite alignment" of tet(M); this terminology could be explained more precisely to aid the general readership

Response: In the revised abstract, we have replaced the term “novel composite alignment” with a more precise and descriptive phrase: “novel composite genetic arrangement flanked by mobile elements, suggesting historical recombination and altered mobility potential.” This change clarifies the genetic context of tet(M), indicating its integration within a composite structure bounded by mobile genetic elements and providing insight into its potential evolutionary origin and mobility. This revision aims to improve clarity for the general readership and to better convey the biological significance of our finding.

Change location: Abstract, lines 37–38.

Comment 2: Discussion: The observed susceptibility to tetracycline in the presence of tet(M) and resistance to clindamycin in the absence of comparable ARGs warrants further mechanistic discussion. Could gene silence, promoter inactivity, or undiscovered resistance factors be involved? Please discuss more about it.

In the revised Discussion (page XX, paragraph XX), we have expanded the explanation to include possible mechanisms for tetracycline susceptibility despite tet(M), such as promoter inactivity, structural mutations, and inducible expression. For clindamycin resistance without erm or lnu genes, we now discuss potential roles of 23S rRNA or ribosomal protein mutations, overexpression of MFS efflux pumps, and uncharacterized determinants such as cfr. These additions provide a clearer rationale for the observed genotype–phenotype discrepancies.

Change location: Discussion, lines 412–431.

Comment 3: Discussion: The discovery of katA and cfa, which are rarely seen in P. acidilactici, is interesting. These findings indicate potential innovative adaptations. Please find literature or brief contextualization to discuss.

Response: we now describe katA as a rare heme-dependent catalase in LAB, potentially acquired from catalase-positive gut or feed bacteria, enhancing oxidative stress tolerance and survival in aerobic and gastrointestinal conditions. Cfa encodes cyclopropane fatty acid synthase, which increases membrane rigidity and hydrophobicity, improving acid and bile tolerance, stability during feed processing, and intestinal persistence. Together, these genes may represent an adaptive signature that enhances resilience for livestock probiotic applications.

Change location: Discussion, lines 457–470.

S1 Fig. Hemolytic Activity Test of five Thai P. acidilactici strains: If you have a clearer image, please replace it for plate marked no. 5 and 6

Response: We have replaced the original S1 Fig. with a high-resolution TIFF image showing plates 5 and 6 more clearly. The updated image improves visibility of hemolysis zones, enabling clearer distinction of alpha-hemolysis. The figure legend has been retained but the image quality now meets PLOS ONE resolution standards (≥300 dpi).

Change location: Supplementary Figure S1.

Editorial Requirement 1: Data Availability Statement

We have revised the Data Availability Statement as requested. The accession numbers for all genome assemblies have been included, along with the repository name (NCBI GenBank) and the link to the database. Additional annotations are provided in the Supplementary File.

Change location: Data Availability Statement, lines 536-542

Editorial Requirement 2: Figure Captions

All figure captions are now included separately in the manuscript following the PLOS ONE format, placed immediately after the first mention of the figure in the text.

Editorial Requirement 3: References

The reference list has been checked and updated to ensure accuracy. No retracted papers are included.

Response to Editorial Requests

PLOS ONE formatting: Manuscript reformatted per PLOS templates.

Data Availability Statement: Revised to include repository names and GenBank accession numbers in the correct format.

Figure captions: Added standalone captions for all figures.

Reference list: Checked for completeness, accuracy, and currency. No retracted articles are cited. New references added for katA, cfa, and resistance mechanism discussion.

We believe these revisions have improved the clarity, completeness, and impact of our manuscript, and we look forward to your further consideration.

On behalf of all authors,

Nuvee Prapasarakul

Corresponding author

---

## [Editor Report · Decision Letter 1]

2 Sep 2025

Integrative genomic characterization of five Pediococcus acidilactici strains reveals differing probiotic safety profiles

PONE-D-25-35195R1

Dear Dr. Prapasarakul,

We’re pleased to inform you that your manuscript has been judged scientifically suitable for publication and will be formally accepted for publication once it meets all outstanding technical requirements.

Kind regards,

Míriam R. García

Academic Editor

PLOS ONE

---

## [Editor Report · Acceptance letter]

PONE-D-25-35195R1

PLOS ONE

Dear Dr. Prapasarakul,

I'm pleased to inform you that your manuscript has been deemed suitable for publication in PLOS ONE. Congratulations! Your manuscript is now being handed over to our production team.

Kind regards,

on behalf of

Dr. Míriam R. García

Academic Editor

PLOS ONE